# Natural variation in the consequences of gene overexpression and its implications for evolutionary trajectories

DeElegant Robinson[1], Michael Place[2], James Hose[3], Adam Jochem[3], Audrey P Gasch[2,3,4]*

[1]Microbiology Doctoral Training Program, University of Wisconsin-Madison, Madison, United States; [2]Great Lakes Bioenergy Research Center, University of Wisconsin-Madison, Madison, United States; [3]Center for Genomic Science Innovation, University of Wisconsin-Madison, Madison, United States; [4]Department of Medical Genetics, University of Wisconsin-Madison, Madison, United States

**Abstract** Copy number variation through gene or chromosome amplification provides a route for rapid phenotypic variation and supports the long-term evolution of gene functions. Although the evolutionary importance of copy-number variation is known, little is understood about how genetic background influences its tolerance. Here, we measured fitness costs of over 4000 overexpressed genes in 15 *Saccharomyces cerevisiae* strains representing different lineages, to explore natural variation in tolerating gene overexpression (OE). Strain-specific effects dominated the fitness costs of gene OE. We report global differences in the consequences of gene OE, independent of the amplified gene, as well as gene-specific effects that were dependent on the genetic background. Natural variation in the response to gene OE could be explained by several models, including strain-specific physiological differences, resource limitations, and regulatory sensitivities. This work provides new insight on how genetic background influences tolerance to gene amplification and the evolutionary trajectories accessible to different backgrounds.

*For correspondence: agasch@wisc.edu

Competing interests: The authors declare that no competing interests exist.

## Introduction

Genetic variation that underlies phenotypic differences provides the material on which evolutionary selection acts. This variation includes single-nucleotide polymorphisms (SNPs), insertions and small deletions, and other structural rearrangements. DNA copy number variants (CNVs) can also serve as a powerful source of variation. CNVs span small tandem duplication of one or few genes to large segmental duplication and even chromosomal aneuploidy that amplifies many genes together (*Hastings et al., 2009*; *Levasseur and Pontarotti, 2011*). Although most duplications are likely lost shortly after creation (*Ohno, 1970*; *Lynch and Conery, 2000*; *Voordeckers and Verstrepen, 2015*), the functional redundancy afforded by gene duplication can support the long-term evolution of new functions (neofunctionalization) or a division of functional labor among the duplicated genes (subfunctionalization) (*Graur and Wh, 2000*). Neutral or nearly neutral variants can accumulate over time in a population, and this standing variation can accelerate evolution when organisms encounter a new environment (*Hermisson and Pennings, 2005*; *Przeworski et al., 2005*; *Barrett and Schluter, 2008*; *Zheng et al., 2020*). But CNVs can also produce immediate changes in cellular fitness due to the effective increase in gene expression, at least for some genes (*Kondrashov, 2012*). For example, yeast cultures challenged by low glucose, sulfate, or nitrogen levels benefit from amplifying genes encoding transporters of glucose (*HXT6/7*), sulfate (*SUL1*), and amino acids (*GAP1*) (*Brown et al., 1998*; *Gresham et al., 2008*; *Gresham et al., 2010*; *Sanchez et al., 2017*). A genome-wide study in *Escherichia coli* showed that 115 amplified genes, including efflux pumps/transporters, regulatory

genes, and prophage genes, increased tolerance to numerous antibiotics and toxins when overexpressed (*Soo et al., 2011*). Consistently, amplification of transporters and resistance genes is an early event in the bacterial evolution of antibiotic resistance (*Sandegren and Andersson, 2009*). Furthermore, whole-chromosome duplication in human fungal pathogens can produce immediate resistance to anti-fungal agents, due to overexpression (OE) of drug efflux pumps and their regulators (*Selmecki et al., 2006*; *Sionov et al., 2010*; *Ni et al., 2013*; *Berman and Krysan, 2020*), and gene amplifications often underlies chemoresistance in cancer cells (*Yasui et al., 2004*; *Mishra and Whetstine, 2016*). These examples illustrate the importance of gene duplication in rapid phenotypic change, especially in response to drugs and environmental stresses where tolerant individuals can rapidly emerge.

While the potential evolutionary benefits afforded by gene amplification are well known, there is also a significant fitness cost (*Adler et al., 2014*; *Moriya, 2015*). The costs and consequences of gene OE are perhaps best studied in *Saccharomyces cerevisiae*. Protein overproduction can cause a shortage of resources, including nucleotides required for additional DNA synthesis, nucleosides consumed by transcriptional burden, and amino acids and ATP required for translation (*Wagner, 2007*). Which resources become limiting can depend on the environment: for example, transcription was shown to be limiting in yeast grown during phosphate starvation, whereas translation is likely limiting when cells are grown in minimal media with low amino acid availability (*Kafri et al., 2016*; *Metzl-Raz et al., 2017*; *Metzl-Raz et al., 2020*). Other cellular processes can become taxed as well. Several studies used the yeast gene-deletion library to investigate genes and processes required to accommodate OE of specific proteins such as GFP (*Farkas et al., 2018*; *Kintaka et al., 2020*). Although results varied somewhat by which protein was overproduced, collectively these studies showed that gene OE can put a burden on mRNA and protein export systems, protein folding chaperones, and protein degradation machinery.

Cells are also impacted by OE of specific genes and functional classes. Several studies have quantified the fitness consequences of gene-OE libraries in laboratory strains of budding yeast *S. cerevisiae* to reveal fitness consequences across many OE genes (*Sopko et al., 2006*; *Ho et al., 2009*; *Magtanong et al., 2011*; *Makanae et al., 2013*). Deleterious OE genes are enriched for those encoding proteins in multi-subunit complexes such as the ribosome. One model to explain this enrichment is that perturbing stoichiometric balances will perturb complex assembly and function (*Papp et al., 2003*; *Veitia et al., 2008*; *Birchler and Veitia, 2012*; *Moriya, 2015*). Cells have mechanisms to control the dosage of some of these proteins, yet high-copy OE can still overwhelm these mechanisms in the cell (*Li et al., 1995*; *Li et al., 1996*; *Fewell and Woolford, 1999*; *Hose et al., 2015*; *Ascencio et al., 2021*). Protein OE can also force promiscuous interactions, perturbing protein interaction networks. Proteins with intrinsically disordered regions (IDRs) are particularly susceptible to promiscuous interactions with other proteins, and deleterious OE genes are enriched for those with IDRs (*Gsponer et al., 2008*; *Vavouri et al., 2009*; *Ma et al., 2010*; *Chakrabortee et al., 2016*). In fact, proteins encoded by gene duplicates fixed in several species are under-enriched for those with IDRs (*Banerjee et al., 2017*), suggesting the impact on long-term evolution. Finally, OE of transcription factors, kinases, and other regulators can trigger broad downstream effects that in turn amplify the expression of other downstream proteins, further taxing proteostasis but potentially also producing phenotypes that could be beneficial (*Sharifpoor et al., 2012*; *Moriya, 2015*; *Youn et al., 2017*). Ultimately, the fitness benefit of gene OE must outweigh the fitness costs in a given environment in order for the duplication to be beneficial to the cell.

Although the evolutionary importance of gene duplication has long been appreciated, little is known about natural variation in the *tolerance* of duplication of specific genes. Variation in the cost of a gene's duplication could have a significant influence on evolutionary trajectories that are accessible to different individuals. Anecdotal evidence shows that different individuals can vary widely in their response to OE of specific genes. For example, prior results from our lab showed that *S. cerevisiae* strains from different genetic lineages have unique fitness responses to overexpressed genes when strains are grown in the presence of toxins (*Sardi et al., 2016*). A major unanswered question is the degree to which reported trends in the fitness consequences of gene OE vary across natural isolates beyond lab strains and how natural variation influences the response to gene OE.

To investigate these questions, we expressed the same high-copy gene OE library applied previously to laboratory *S. cerevisiae* strains, in 15 different yeast isolates together representing four lineages and several admixed strains, to explore the variation in tolerance to gene OE. Our results

distinguish universal effects common to many studied strains versus strain-specific effects, including global responses independent of the OE gene as well as gene-specific sensitivities. We present evidence for several general models explaining strain-specific variation in the response to gene OE. These results raise important implications for the accessibility of evolutionary trajectories afforded by gene OE depending on genetic background.

## Results

### Overview

We chose 15 genetically diverse *S. cerevisiae* strains for analysis, including strains from four defined genetic lineages (including European Wine, North American Oak, Asian, and West African), one commonly used lab strain (BY4743), and three strains that represent recently admixed 'mosaic' strains. These isolates were collected from diverse environments including soil, vineyards, sake production, sewage, and clinical samples (*Capriotti, 1955*; *Gerke et al., 2006*; *Kurtzman, 1986*; *McCullough et al., 1998*; *Sniegowski et al., 2002*; *Strope et al., 2015*; *Supplementary file 1*). In addition to genetic diversity, these strains display extensive phenotype variation, for example, in nitrogen and carbon utilization (*Warringer et al., 2011*), nutrient requirements (*Liti et al., 2009*; *Warringer et al., 2011*), and stress tolerance (*Kvitek et al., 2008*; *Liti et al., 2009*; *Will et al., 2010*; *Warringer et al., 2011*; *Strope et al., 2015*; *Zheng and Wang, 2015*; *Sardi et al., 2016*; *Sardi et al., 2018*).

Each strain was transformed with the MoBY 2.0 library that includes ~4900 open reading frames (ORFs) with their native upstream and downstream sequences, cloned into a high-copy 2-μm replicating plasmid (*Ho et al., 2009*; *Magtanong et al., 2011*). We chose the high-copy expression system to expose gene-specific fitness differences that may be too subtle to score when genes are merely duplicated. Although the lab strain replicates the empty vector at ~11 copies per haploid genome, most other strains maintain ~2–5 copies per haploid genome (see *Figure 2—figure supplement 2*). All strains were readily transformed with the library and grew at expected growth rates in selective media, indicating that all strains could maintain the plasmid. An aliquot of each library transformed culture was collected before and after 10 generations of competitive growth, and relative plasmid abundance was scored by quantitative sequencing of plasmid barcodes to measure changes in plasmid abundance in the population, in biological triplicate (*Figure 1A*, see Materials and methods).

Barcode abundance was normalized to the total number of reads per sample, thus producing a fitness score relative to the total set of genes expressed in each strain and accounting for strain-specific differences in library expression. We measured the $\log_2$(fold change) in relative barcode abundance after 10 generations of competitive growth, which we refer to as the relative fitness score. Genes that are detrimental when overexpressed will drop in frequency in the population because of reduced cell growth or because cells suppress the abundance of toxic plasmids (*Makanae et al., 2013*), both of which we interpret as a relative fitness defect. In contrast, beneficial plasmids will rise in frequency in the population over time. We focus on genes with a significant fitness effect when OE at a false discovery rate (FDR)<5% (see Materials and methods, *Supplementary file 4*).

We first validated our results by comparing fitness effects measured in lab strain BY4743 to a previous study using a similar library of yeast genes expressed from their native promoters in a similar strain background (*Makanae et al., 2013*). There was highly significant overlap between the 851 genes, we identified that produce a defect upon OE and previous results of Makanae et al. (p=8 $\times 10^{-45}$, Hypergeometric test) despite differences in media and experimental conditions (*Makanae et al., 2013*). Deleterious OE genes identified in both studies were enriched for genes involved in translation, including ribosomal proteins and essential genes, and a largely overlapping group of genes repressed as part of the yeast Environmental Stress Response (*Gasch et al., 2000*) (p<1.4$\times 10^{-10}$, Hypergeometric test). Thus, our approach is robust to replication and comparable to previous studies, validating our methods.

We next quantified the library effects across strains (*Figure 1B*). We identified 4064 genes whose OE produced a reproducible fitness effect in at least one strain (FDR<0.05), with a median of 1726 genes per strain. However, there was a wide range in the number of consequential genes (*Figure 2A*). Mosaic strain Y2209 was affected by 635 OE genes, whereas Y12 (isolated from African



**Figure 1.** Overview of experiment and results. (**A**) Isolates transformed with the MoBY 2.0 overexpression library were grown competitively and changes in plasmid abundance were quantified, see Materials and methods for details. (**B**) Heat map of hierarchically clustered log₂(relative fitness scores) for 4064 genes (rows) measured in 15 strains in biological triplicate (columns) after 10 generations of growth. Strain labels are colored according to lineage. Blue and yellow colors represent plasmids that become enriched or depleted in frequency to indicate fitness defects or benefits, respectively, according to the key. Some barcodes with missing values after growth were inferred (see Materials and methods); those that are significant are indicated as an orange box in the heat map. A source data file is included (see *Figure 1—source data 1*: Hierarchical clustered fitness scores). The online version of this article includes the following source data for figure 1:

**Source data 1.** Hierarchical clustered fitness scores.

palm wine but genotypically similar to Asian strains) was affected by 3060 OE genes. (We note that the low number of genes identified in YPS606 may be influenced by reduced statistical power since only duplicates of that strain were analyzed, and the lower abundance of the 2-μm plasmid in the case of YJM1592 and YJM978.) Most significant OE genes were detrimental, although there were some differences across strains. For example, whereas roughly half of the significant OE genes in the lab strain BY4743 caused a defect, over 95% of significant OE genes in the Y12 strain were detrimental (*Figure 2—figure supplement 1*).

In addition to the variable number of OE genes with fitness consequences, strains also varied in the severity of the defects (*Figure 2B*). While the median fitness cost of deleterious OE genes was not correlated overall with the number of deleterious genes per strain, strains with the most deleterious genes (NCYC3290, YJM1389, and Y12) did show an expanded range of fitness costs, with more genes showing very strong deleterious effects compared to other strains (*Figure 2B*). Importantly, there was no correlation between the number of deleterious OE genes and Moby 2.0 copy number maintained in the strains (*Figure 2—figure supplement 2A*), as expected since our normalization procedure reflects gene fitness effects relative to the overall library in that strain. Most significant genes produced fitness effects in only a subset of strains (*Figure 2C*), even though results were highly reproducible within strain replicates, with over half the 4064 significant genes producing a



**Figure 2.** Strain backgrounds display a wide range of fitness effects. (A) The number of deleterious genes in each strain (FDR<0.05), colored by lineage as in *Figure 1B*. The number of deleterious and beneficial genes in each strain (FDR<0.05) are shown in *Figure 2—figure supplement 1*. The number of deleterious genes per strain is not related to 2-μm abundance (see *Figure 2—figure supplement 2*). (B) The distributions of log₂ (relative fitness scores) for genes identified as deleterious in each strain. Imputed ratios were not included. The strains are ordered based on the number of fitness defects from smallest to largest (left to right); YPS606 (asterisk) likely had lower statistical power due to analysis of only duplicates. (C) Deleterious genes were binned according to the number of strains in which the gene had a deleterious fitness effect (x-axis). Commonly deleterious genes were defined as a set of 431 genes with a deleterious effect in ≥10 strains.

The online version of this article includes the following figure supplement(s) for figure 2:

**Figure supplement 1.** Number of significant genes across strains.

**Figure supplement 2.** The number of deleterious genes per strain is not related to 2-μm abundance.

defect in only four or fewer strains. Together, these results highlight that different strains respond differently to gene OE, on both broad and gene-specific scales.

## Genes whose overexpression is deleterious across many strains are functionally related

Before investigating strain-specific effects, we first characterized genes producing a defect in many strains (*Figure 3*). There were 431 OE genes that produced a significant defect in at least 66% of strains (FDR<0.05), and we refer to these as 'commonly deleterious' OE genes. This set was heavily enriched for genes involved in translation including ribosomal proteins, ribosome biogenesis factors, and other genes repressed during stress in the Environmental Stress Response. The group was also enriched for genes encoding helicases and ATP binding proteins, mitosis regulators, proteins that

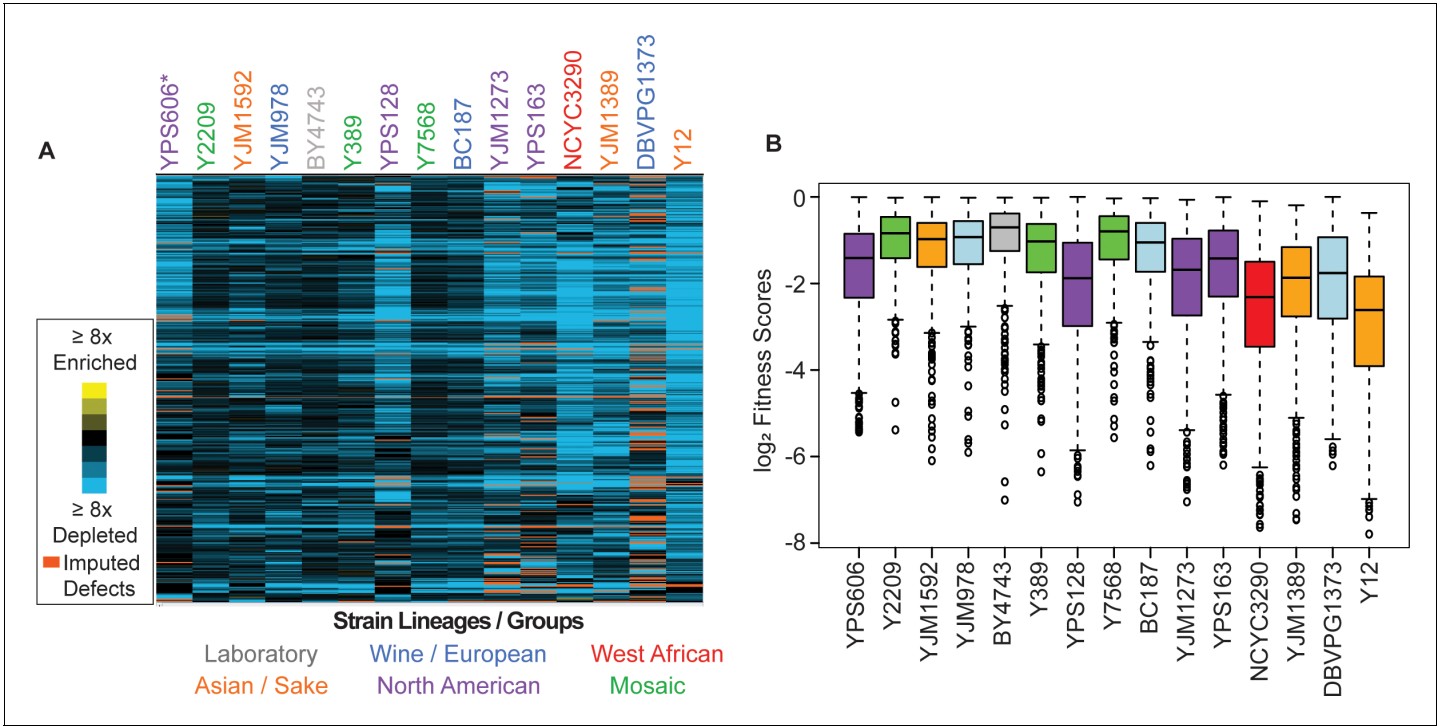

**Figure 3.** Commonly deleterious genes affect many strains but to different degrees. (**A**) Heat map of log₂ (relative fitness scores) as shown in *Figure 1B* but for 431 commonly deleterious genes. (**B**) The distribution of the log₂ (relative fitness scores) (taking the replicate average for each gene) for 431 commonly deleterious genes are plotted. Imputed scores were not included. The strains are ordered based on the total number of deleterious genes, from smallest to largest (left to right). A source data file is included (see *Figure 3—source data 1*).

The online version of this article includes the following source data for figure 3:

**Source data 1.** Hierarchical clustered commonly deleterious genes fitness scores.

localize to the nucleus, and essential proteins (p<1E−4, Hypergeometric test). All of these categories remained significant if genes involved in translation were removed from the analysis (see Materials and methods), demonstrating that the enrichments are not due to overlap with translation factors.

The Balance Hypothesis (*Birchler and Veitia, 2012*) posits that genes encoding proteins in multi-subunit complexes or with many protein-protein interactions cause stoichiometric imbalances and thus toxicity when overexpressed. Indeed, we found that commonly deleterious genes had more protein interactions (*Oughtred et al., 2021*) (p=4.0×10⁻⁶⁹, Wilcoxon test) and included more proteins that form complexes as defined by *Pu et al., 2009* (p=6.6×10⁻¹², Hypergeometric test) compared to all other genes measured in at least one strain. This confirms the result of Makanae et al. who also found that dosage-sensitive genes in the lab strain are enriched for proteins with many interactions and proteins in complexes (*Makanae et al., 2013*). Translation factors accounted for 163/431 (~38%) of the commonly deleterious genes and are known to participate in many interactions, raising the possibility that this functional group is driving the result. Even after removing genes involved in translation from consideration (see *Supplementary file 2*), the remaining commonly deleterious genes were still enriched for complex members and proteins with many physical interactions, strongly suggesting these features as driving factors in toxicity. We also noted that common deleterious OE genes contain a higher proportion of disordered proteins (as estimated by median IUPRED scores for each protein; *Mészáros et al., 2018*) than other genes measured in the experiment (p<4.7×10⁻¹², Wilcoxon test), which is consistent with other analyses of deleterious OE genes (*Vavouri et al., 2009*; *Ma et al., 2010*).

Another hypothesis for dosage sensitivity is that already abundant proteins emerging from genes and transcripts that are highly transcribed and/or translated may be more subject to aggregation if further overexpressed. While the total set of common genes is expressed at higher mRNA and protein abundance, this trend was driven by translation factors and was not significant when translation factors were removed from the analysis. Thus, while high abundance of translation factors could contribute to their dosage effects, high expression alone is not a predictor of OE toxicity for other genes. Together, our work suggests that the propensity to disrupt protein interaction networks is likely a driving factor in OE gene toxicity.

## Strain-specific responses to library overexpression may reflect global differences in resource allocation

Although we identified a common set of deleterious OE genes, strains clearly varied in their response to the library, in multiple distinct ways. Even for commonly deleterious genes, isolates varied in the severity of their responsiveness (*Figure 3*). In general, strains with a larger number of deleterious OE genes displayed more severe median relative fitness costs in response to common-gene OE than strains with fewer deleterious OE genes—the exception was North American oak-soil strains that showed aberrantly high fitness costs of common-gene OE even though they were not the most sensitive in terms of number of deleterious-gene effects (*Figure 3B*, purple boxes). These differences cannot be explained by gross differences in gene-OE levels, since strains with comparable Moby 2.0 copy numbers showed vastly different responses. For example, BC187 and YPS128 carry comparable plasmid copy numbers (*Figure 2—figure supplement 2A*), yet YPS128 is much more sensitive to commonly deleterious genes than BC187. Instead, these data suggest that some strains are more sensitive to gene amplification, even for OE genes that are commonly deleterious across many strains.

Several possibilities could explain these results. One is that cells have different capacities for tolerating protein overproduction, regardless of the OE gene. To test this, we measured growth rates in response to OE of a non-native yeast protein, GFP, expressed from the highly active *TEF1* promoter on a low-copy CEN vector. Strains with the most deleterious OE genes in fact did not show higher sensitivity to GFP overexpression; however, three of the four North American oak-soil strains did, as indicated by significantly slower doubling times (*Figure 4A*). The reduced growth was not due to excessive GFP production as the oak strains express GFP at levels comparable to other strains tested (*Figure 4—figure supplement 1A*). One possibility is that these strains have reduced capacity to tolerate high protein production due to general amino acid shortage. To test, we measured growth rates during GFP OE when strains were grown in synthetic media with and without amino acids; however, growth rates of the oak strains as a group were not different than other strains, all of which grew comparably slower in the absence of amino acids (*Figure 4B*). The sensitivity of all strains to amino acid shortage is consistent with previous reports in the lab strain (*Farkas et al., 2018*). However, the degree of sensitivity to amino acid shortage did not correlate with the overall number of deleterious genes per strain, indicating that this is unlikely a driving factor explaining strain-specific effects.

Another possibility is that strains vary in the burden of protein overproduction in the context of the 2-μm plasmid, which may create a different type of stress on some strains. There was no overall correlation between the number of deleterious OE genes and the abundance of the strain's native 2-μm plasmid (p>0.14) (although some strains with very low native 2-μm abundance also had a high number of deleterious genes [*Figure 2—figure supplement 2B*]). There was also no correlation between deleterious gene number and Moby 2.0 copy number. Nonetheless, we wondered if some strains may be more sensitive to the burden of the Moby 2.0 plasmid.

To test this, we measured growth rates of strains with and without the Moby 2.0 empty vector. Although many strains grew slower when expressing the empty vector under selection, some strains were more significantly affected (*Figure 4C*). While none of the tests passed an FDR<0.05, the trend was consistent across replicates for most strains. Indeed, the number of deleterious OE genes was correlated with the percent decrease in growth rate when strains expressed the empty Moby 2.0 vector (r=0.7, p=0.005, *Figure 4D*). Interestingly, strains with the greatest sensitivity to the empty Moby 2.0 vector were not the same as those most sensitive to GFP overproduction, revealing separable effects. Thus, the added stress of DNA/Moby 2.0 overproduction may render some strains more sensitive to gene OE (see Discussion).



**Figure 4.** Strains show different sensitivities to protein and DNA expression. (A) The average and standard deviation (n=3) of doubling times for strains containing a CEN empty vector (light gray) or CEN vector expressing GFP from the *TEF1* promoter (purple). Asterisk indicates FDR<0.005 and plus sign indicates FDR<0.07, paired t-tests. Western blot analysis of strains expressing the GFP vector is shown in *Figure 4—figure supplement 1*. (B) Doubling times of GFP-expressing strains from (A) grown in synthetic medium without amino acids relative to synthetic-complete medium (n=3). All strains grow significantly slower without amino acids (asterisk; FDR<0.05). (C) Average and standard deviation of doubling times for each strain carrying the Moby 2.0 empty vector grown in selection (blue) or with no vector and grown in the absence of selection (gray) (n=3). A plus sign indicates FDR<0.1, one-tailed t-test. (D) Number of deleterious genes per strain (y-axis) compared to the % decrease in growth rate for each strain carrying the Moby 2.0 empty vector (x-axis) as measured in (C). There is a positive correlation between the number of deleterious OE genes and % decrease in doubling time in response to the vector (r=0.7, p=0.005, excluding YPS606). FDR, false discovery rate.

The online version of this article includes the following source data and figure supplement(s) for figure 4:

**Figure supplement 1.** Western blot analysis of anti-GFP (red) and anti-PGK1 loading control (green) in strains carrying to GFP plasmid, grown to log phase in SC.

**Figure supplement 1—source data 1.** Raw blot detecting anti-PGK1 in samples 1–13.

**Figure supplement 1—source data 2.** Raw blot detecting anti-PGK1 in samples 14–16.

**Figure supplement 1—source data 3.** Raw blot detecting anti-GFP in samples 1–13.

**Figure supplement 1—source data 4.** Raw blot detecting anti-GFP in samples 14–16.

**Figure supplement 1—source data 5.** Uncropped Western blot images for all samples.

## Strain-specific responses to specific genes implicate models for variable OE tolerance

In addition to gene-independent differences in tolerating overproduction, strains also varied in their response to specific OE genes. Most overexpressed genes produced a relative fitness effect in a small number of strains, suggesting the widespread influence of genetic background (*Figure 2C*). To further investigate, we identified genes whose OE produced a significant fitness effect in each strain and not more than two others, which we defined as 'strain-specific' gene lists. The lists ranged from 41 genes in the Y2209 strain to 1763 genes in the Y12 strain (*Figure 5A*). It is important to note that some of these effects in strains overly sensitive to the empty vector could still reflect generalized strain sensitivities. For example, 60% of the deleterious genes identified by our criteria as 'strain-specific' in Y12 were shared with another of the top four strains most sensitive to the empty vector (DVBPG1373, YJM1592, YPS163, and YJM1389, *Figure 4D*). Thus, some of the identified genes may be deleterious if OE in other strains growing in suboptimal or stressful conditions.

Next, we investigated functional or biophysical features enriched in each strain's list, which might implicate strain-specific constraints in tolerating gene OE. We compared, separately, genes that were specifically beneficial or detrimental in a given strain to a list of genes in that strain that were well-measured but produced no effect on fitness (FDR>0.1, see Materials and methods). To interpret the results, we also measured strain-specific gene expression differences through triplicated RNA-seq transcriptomic experiments (*Supplementary file 5*) to explore connections between native gene expression and the fitness consequences of gene OE.

Among the many sets of functional and biophysical enrichments (see *Supplementary file 6*), several themes emerged that suggest models for strain-specific responses to gene OE. The first model may be particular to our experimental design, in which the S288c allele is expressed in the library. Beneficial OE genes in BC187 and Y7568 harbored an overabundance of nonsynonymous SNPs between the overexpressed S288c allele and the strains' native allele (p<0.0009, Wilcoxon test). One possibility is that the S288c allele may complement deleterious SNPs accumulated in those strains. For two other strains, YJM1273 and Y7568, deleterious OE genes showed a higher proportion of amino acid differences between native and expressed alleles (p<$2\times10^{-4}$, Wilcoxon test). Here, allelic conflict could explain strain-specific sensitivity to the S288c allele, if the focal strain evolved its own polymorphisms. These strains did not show higher genetic distances overall from S288c, raising the possibility that the high rate of allelic differences may reflect genes under accelerated evolution.

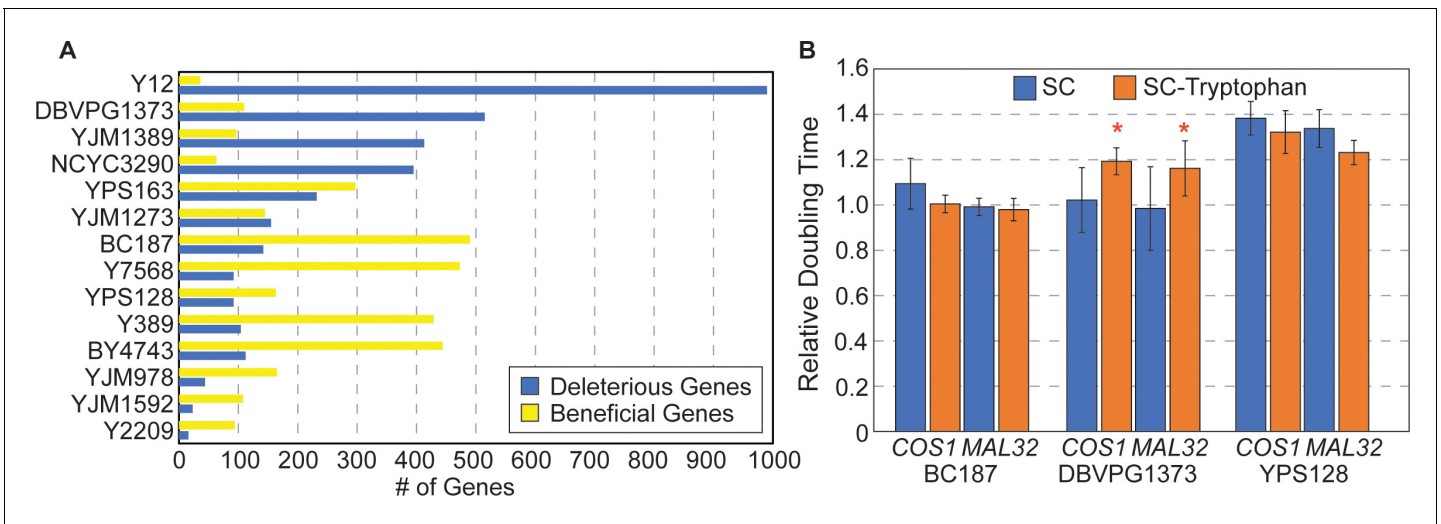

**Figure 5.** Strain-specific responses to gene OE. (**A**) The number of strain-specific genes that were deleterious (blue) or beneficial (yellow) are shown for each strain. (Note: YPS606 was not included due to low statistical power of duplicate replicates.) (**B**) Average and standard deviation of doubling times in denoted strains overexpressing *COS1* or *MAL32*, relative to the growth rate of each strain carrying the empty vector, when strains were grown in synthetic complete medium (blue) or medium lacking tryptophan (orange) (n=3). Asterisk indicates slower relative growth in tryptophan-minus media (p<0.05, one-tailed paired t-test).

A second model explaining strain-specific gene OE effects is one in which the unique physiology of a strain causes a unique response to gene OE. This model is supported by functional enrichments for strain-specific OE gene lists, especially when those functions relate to differentially expressed genes in the same strains (*Supplementary file 7*). There were several functional categories that were enriched for multiple strain-specific lists. For example, OE genes beneficial to DBVPG1373 or Y12 were enriched for genes involved in the mitotic cell cycle (p<0.0004, Hypergeometric test). Both strains showed differential gene expression of cell-cycle genes: G2/M genes were expressed significantly lower in both strains and, in the case of DBVPG1373, S-phase genes were expressed significantly higher, raising the possibility of underlying differences in the strains' cell-cycle regulation or timing. Another recurring example is reflected in nuclear-encoded mitochondrial genes. Beneficial OE genes affecting BC187 and Y389 were enriched for mitochondrial functions including respiration and cytochrome complex assembly, respectively; both strains showed altered expression of genes related to mitochondria. In contrast, genes whose OE was deleterious to Y12 were enriched for those encoding proteins in the mitochondrial matrix, and this functional category was enriched among genes expressed higher in Y12. Notably, genes related to mitochondrial function were also deleterious in a number of other strains sensitive to the empty vector (*Figure 4D*), raising the possibility that these genes may be generally deleterious in suboptimal or stressful conditions (or conversely that strains with inherent differences in mitochondrial function, suggested by the differential expression in the native strains, are sensitive to the vector). Although the exact connections in these cases will require experimentation to elucidate, that related functional categories are associated with a strain's unique gene-OE susceptibilities and their unique expression differences points to physiological differences that could influence strain responses (see Discussion).

A third model is a specific example of physiological differences—strain-specific resource limitation. OE genes deleterious to vineyard strain DBVPG1373 encode proteins with a higher proportion of tryptophan compared to inconsequential genes (p=8.1×10$^{-5}$, Wilcoxon test). That the encoded proteins are related by their composition hinted that limited tryptophan availability in this strain could sensitize cells to proteins high in tryptophan content. Interestingly, transcriptomic profiling revealed that genes repressed in this strain are enriched for genes encoding aromatic amino acid biosynthesis proteins (p=4.1×10$^{-6}$, Hypergeometric test), and other amino acids. Tryptophan is a precursor for de novo synthesis of NAD+, and other genes in this pathway (*BNA1, 4, 6,* and *7*) as well as genes in the nicotinic acid/nicotinamide salvage pathway (*NTP1, NRK1,* and *TRP2*) were also repressed in DBVPG1373. Together, these data raised the possibility that DBVPG1373 is sensitive to conditions that deplete tryptophan from the cell. One prediction is that DBVPG1373, but not other strains, should be especially sensitive to OE of tryptophan-containing proteins when grown in tryptophan-limiting media. To test this, we compared growth rates of DBVPG1373 and control strains BC187 and YPS128 overexpressing tryptophan-enriched genes *COS1* (3.9% tryptophan) or *MAL32* (3.4% tryptophan) in synthetic media with and without tryptophan. DBVPG1373 expressing Moby 2.0 vectors grew especially poorly in synthetic media compared to the other strains, perhaps obscuring the deleterious effects of *COS1* and *MAL32* OE (*Figure 5B*). Nonetheless, when overexpressing the genes, DBVPG1373 was reproducibly more sensitive in the absence of tryptophan (p≤0.05, one-tailed paired t-test), whereas the other strains were not. It is possible that this strain is more sensitive to any OE genes under these conditions, if the cellular system is taxed in the absence of tryptophan. Nonetheless, these results support the model that strains can vary in resource limitation (see Discussion).

A final model is that OE of regulators can perturb networks of downstream proteins, and strains sensitive to those networks will be overly sensitive to OE of the regulators. One possible example is seen in mosaic strain Y7568, whose deleterious OE gene list is enriched for DNA binding proteins (p=1.8×10$^{-4}$, Hypergeometric test), including site-specific transcription factors (Flo8, War1, Pho2, Pdr1, Pdr8, and Hcm1). Collectively, the combined set of these factors' targets is heavily enriched for genes encoding plasma membrane-localized proteins including drug and other transporters (p=5.7×10$^{-5}$). Remarkably, the list of OE genes that are deleterious in Y7568 is also enriched for genes encoding plasma membrane proteins (p=1.8×10$^{-4}$). Although not enriched above chance, 9% of the deleterious OE genes in Y7568 are direct targets of one of the six factors above (*Monteiro et al., 2020*). These connections give support to the model that OE of regulators can be deleterious via OE of downstream toxic genes, and that the effects can be strain-specific.

## Genes highly beneficial to some strains may relate to 2-μm replication

We identified a unique cluster of 21 genes whose OE was strongly beneficial in over half (60%) of the strains, but notably not the lab strain. Although not enriched for any specific functions, the group included multiple genes involved in ribosome biogenesis/function (*RRP6*, *RRP7*, *LOC1*, *RPL35B*, and *DOM34*). Interestingly, over half the genes are located next to a centromere, and on closer inspection the plasmids would have cloned the centromere in the genes' upstream regions. This was interesting because 2-μm segregation is closely coupled with chromosome segregation (*Liu et al., 2014*; *Mehta et al., 2002*). Past work showed that cloning centromeric sequences onto a 2-μm replicating plasmid reduces copy number to that of chromosome levels (*Apostol and Greer, 1988*). This raised the possibility that CEN sequences rather than the cloned genes could influence fitness effects in the cell, at least for a subset of these genes.

We selected two plasmids from the beneficial gene cluster that encode *ERG26* (involved in ergosterol biosynthesis) or *LOC1* (involved in mRNA localization and also ribosome biogenesis) and the adjacent CEN encompassed in their upstream regions. Oak-soil strain YPS128 carrying the *ERG26* or *LOC1* plasmids grew faster than the empty vector control, confirming the fitness benefit to this strain (*Figure 6A*). If the reduced copy number and growth benefit afforded to YPS128 is due only to the cloned CEN, then deleting the entire ORF or the start codon should retain the benefits provided by the CEN that remains on the plasmid. We generated derivatives of each plasmid in which the ORF (but not upstream sequence) was deleted or the start codon replaced with a stop codon (M*). Although the plasmid in which *ERG26* was deleted showed some benefits, the other mutants did not, even though the *ERG26* M* variant retained lower copy number (*Figure 6*). Thus, although half the plasmids in this cluster had cloned the CEN, the gene product may still be important. While future research will be required to disentangle why these plasmids provide a benefit, these results are yet another example in which strains respond differently to the same experimental environment.

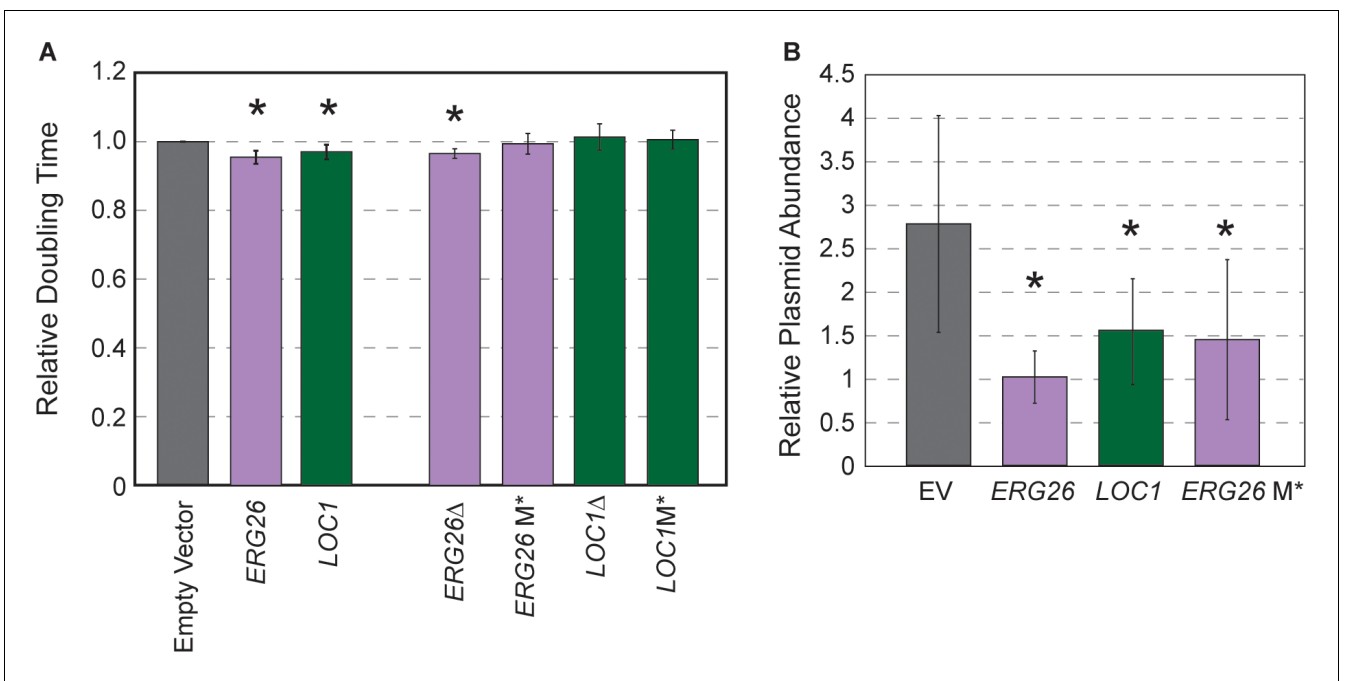

**Figure 6.** Highly beneficial genes may relate to 2-μm replication. (**A**) Average and standard deviation of doubling times of strain YPS128 carrying *ERG26* or *LOC1* Moby 2.0 vectors relative to the empty vector (n=5), or vectors in which the gene portion was deleted (Δ) or the start codon was mutated (M*). Asterisk indicates significantly faster growth rate versus the empty-vector control (p<0.05, paired t-test). (**B**) Abundance of Moby 2.0 vectors was adjusted relative to a CEN vector (see Materials and methods, n=5 except for *LOC1* in which n=3).

## Discussion

Our work shows that genetic background has a profound influence on how cells respond to gene OE. Out of the ~4000 genes whose OE impacted fitness in at least one isolate, only ~12% influenced fitness in 10 or more of the 15 strains. A hallmark of the 431 commonly deleterious OE genes is their potential to perturb protein-interaction networks, either because the proteins naturally display many connections or because their biophysical properties may force promiscuous interactions when over-expressed (*Gsponer et al., 2008*; *Vavouri et al., 2009*; *Ma et al., 2010*; *Chakrabortee et al., 2016*). But even for commonly deleterious genes, strains varied in the cost incurred by their OE. We suggest two main classes for strain-specific effects. One is general responses that may be independent of the overexpressed gene. For example, some strains became sensitized to gene OE in the context of the high-copy plasmid: those with greater sensitivity to the empty vector (independent of the vector copy number) showed proportionately more deleterious OE genes and with greater fitness costs. Whether this limitation is due to the burden of extra DNA or something related to the stress of 2-μm replication is not clear; nonetheless, the result unmasks strain-specific vulnerabilities that have a broad impact. We note that many of the genes scored as deleterious in these sensitive strains may cause fitness defects in other strains grown in suboptimal or stressful conditions. The second class of strain-specific effects pertains to gene-specific responses. Our results suggest several explanatory models, including strain-specific physiological differences, strain-specific resource limitation, and unique sensitivities to network perturbation by regulator amplification.

The implications of our study are several-fold. The first is that strains may have differential access to evolutionary trajectories if the cost of gene duplication varies across individuals. CNVs can produce immediate phenotypic gains for genes not subject to dosage control (*Zhang et al., 2009*; *Kondrashov, 2012*; *Hose et al., 2015*), but they can also produce standing genetic variation on which selection can later act. This standing variation is important for the long-term functional evolution (*Ohno, 1970*; *Graur and Wh, 2000*) and it can also accelerate evolution when selective pressures change (*Zheng et al., 2020*). If the cost of gene duplication, or simply increasing expression from a single-copy gene, is higher in some backgrounds, then those strains may be less likely to evolve through CNV mechanisms. An extreme example is whole-chromosomal aneuploidy, which is a potent mode of rapid evolution that is prevalent in some genetic backgrounds yet poorly tolerated in others (*Torres et al., 2007*; *Gallone et al., 2018*; *Hose et al., 2020*; *Scopel et al., 2021*). Differences in aneuploidy tolerance could be heavily influenced by different gene-specific sensitivities across strains. Consistent with the notion that these differences can affect evolutionary trajectories, several studies have found that different fungal strains evolve through different mechanisms when exposed to the same laboratory selections, where some genetic backgrounds leverage aneuploidy, polyploidy, and CNV while others do not (*Filteau et al., 2015*; *Gerstein and Berman, 2020*; *Tung et al., 2021*).

Another implication of our work is the interplay between gene OE, genetic background, and environment. Past work has shown that the cost of gene OE in a single strain can vary with nutrient limitation (*Wagner, 2005*; *Kafri et al., 2016*; *Frumkin et al., 2017*; *Farkas et al., 2018*; *Kintaka et al., 2020*). We propose that variation in environmental responses will further reveal variation in gene OE differences across strains, as hinted at by our studies. For example, strain Y7568 showed little sensitivity to GFP OE—unless amino acids were removed from the media in which case it grew among the worst across strains (*Figure 4*). The sensitivity of vineyard strain DBVPG1373 to tryptophan-containing proteins was exacerbated by tryptophan depletion, an environment that produced little added effect on other strains. Even the sensitivity to the Moby 2.0 empty vector may have unmasked strain sensitivities that are not evident if those strains are grown in other environments. Understanding gene-by-environment interactions is among the greatest challenges in genetics. Understanding how this interplay influences evolutionary potential is even more complicated but beginning to emerge through experimental studies (*Filteau et al., 2015*; *Tung et al., 2021*).

The results of our work also have broad application, from microbial engineering to human health. Many industrial processes use gene OE to improve microbial traits (*Keasling, 1999*; *Xie and Fussenegger, 2018*). Understanding (and ultimately predicting) how the response to engineering strategies will vary across host strains could accelerate engineering efforts (*Steensels et al., 2014*; *Sardi and Gasch, 2017*; *Sardi and Gasch, 2018*). Interpreting functional variants, from SNPs to CNVs, is also a major goal in human genetics and precision medicine. While already a colossal goal,

incorporating genetic background interactions in such predictions will be fundamental. Elucidating mechanistic underpinnings in model organisms will continue to pave the way toward deeper understanding.

# Materials and methods

## Key resources table

| Reagent type (species) or resource | Designation | Source or reference | Identifiers | Additional information |
|---|---|---|---|---|
| Gene (kanʳ) | kanʳ | Yeast knockout Collection; Horizon Discovery | | kanMX |
| Strain, strain background (*Saccharomyces cerevisiae*) | BY4743 MATa/α his3Δ1/his3Δ1 leu2Δ0/leu2Δ0 LYS2/lys2Δ0 met15Δ0/MET15 ura3Δ0/ura3Δ0 | ATCC | BY4743 | |
| Strain, strain background (*S. cerevisiae*) | BC187 | *Gerke et al., 2006*, doi: 10.1534/genetics.106.058453 | BC187 | |
| Strain, strain background (*S. cerevisiae*) | DBVPG1373 | *Capriotti, 1955* | DBVPG1373 | |
| Strain, strain background (*S. cerevisiae*) | NCYC3290 | Bili wine, *Liti et al., 2009*, doi: 10.1038/nature07743 | NCYC3290 | |
| Strain, strain background (*S. cerevisiae*) | Y12 | Palm wine, C. Kurtzman and the ARS culture collection | Y12 | |
| Strain, strain background (*S. cerevisiae*) | Y2209 | *Lepidopterous sample, C. Kurtzman and the ARS culture collection* | Y2209 | |
| Strain, strain background (*S. cerevisiae*) | Y389 | Mushrooms, C. Kurtzman and the ARS culture collection | Y389 | |
| Strain, strain background (*S. cerevisiae*) | Y7568 | Papaya, C. Kurtzman and the ARS culture collection | Y7568 | |
| Strain, strain background (*S. cerevisiae*) | YJM1273 | *Sniegowski et al., 2002*. doi: 10.1111/j.1567–1364.2002.tb00048.x | YJM1273 | |
| Strain, strain background (*S. cerevisiae*) | YJM1389 | *Strope et al., 2015* doi: 10.1101/gr.185538.114 | YJM1389 | |
| Strain, strain background (*S. cerevisiae*) | YJM1592 | *Strope et al., 2015* doi: 10.1101/gr.185538.114 | YJM1592 | |

*Continued on next page*

*Continued*

| Reagent type (species) or resource | Designation | Source or reference | Identifiers | Additional information |
|---|---|---|---|---|
| Strain, strain background (*S. cerevisiae*) | YJM978 | Human, clinical, *Strope et al., 2015* doi: 10.1101/gr. 185538.114 | YJM978 | |
| Strain, strain background (*S. cerevisiae*) | YPS128 | *Sniegowski et al., 2002*. doi: 10.1111/j.1567–1364.2002.tb00048.x | YPS128 | |
| Strain, strain background (*S. cerevisiae*) | YPS163 | *Sniegowski et al., 2002*. doi: 10.1111/j.1567–1364.2002.tb00048.x | YPS163 | |
| Strain, strain background (*S. cerevisiae*) | YPS606 | Oak tree bark, *Sniegowski et al., 2002*. doi: 10.1111/j.1567–1364.2002.tb00048.x | YPS606 | |
| Antibody | Rabbit anti-GFP (Rabbit polyclonal) | Abcam | Abcam Cat# ab290, RRID:AB_303395 | (1:2000) |
| Antibody | Mouse anti-PGK1 (Mouse monoclonal) | Abcam | Abcam Cat# ab113687, RRID:AB_10861977 | (1:1000) |
| Recombinant DNA reagent | Moby 2.0 yeast gene overexpression library | *Magtanong et al., 2011*. doi: 10.1038/nbt. 1855 | | |
| Recombinant DNA reagent | pPKI | *Hose et al., 2020* DOI: 10.7554/eLife. 52063 | AGB185 | CEN plasmid with the natMX selection marker |
| Recombinant DNA reagent | pJH2 | This study | AGB91 | pKI's NAT cassette was replaced with the KAN cassette |
| Recombinant DNA reagent | pJH3 | This study | AGB92 | GFP was digested out of pJH2 to obtain pJH3; this was used as the CEN empty vector control |
| Recombinant DNA reagent | pJH2_TEFprom-GFP-ADH1term | This study | In BY4741 under AGY1566 | pJH2 plasmid with TEF promoter and ADH1 terminator sewn together with GFP |
| Recombinant DNA reagent | MoBY 2.0 Empty Vector Control | *Magtanong et al., 2011*. doi: 10.1038/nbt. 1855 | AGB181 | |
| Recombinant DNA reagent | ERG26Δ | This study | AGY1672 | Strain expressing plasmid from Moby 2.0 with ERG26 coding sequence deleted |
| Recombinant DNA reagent | ERG26 M* | This study | AGY1673 | Strain expressing ERG26 plasmid from Moby 2.0 with start codon replaced with a stop codon |
| Recombinant DNA reagent | LOC1Δ | This study | AGY1674 | Strain expressing plasmid from Moby 2.0 with LOC1 coding sequence deleted |

*Continued*

| Reagent type (species) or resource | Designation | Source or reference | Identifiers | Additional information |
| --- | --- | --- | --- | --- |
| Recombinant DNA reagent | LOC1 M* | This study | AGY1675 | Strain expressing LOC1 plasmid from Moby 2.0 with start codon replaced with a stop codon |
| Chemical compound, drug | G418 (G-418 Disulfate) | RPI | RPI SKU G64000 | CAS #108321-42-2 |

## Strains and growth conditions

Strains used in this study are listed in *Supplementary file 1*. Unless otherwise indicated, strains were grown in rich YPD medium (10 g/L yeast extract, 20 g/L peptone, 20 g/L dextrose) in shake flasks at 30°C. Each strain was transformed with a pool of the molecular barcoded yeast ORF library (MoBY 2.0) containing 4871 pooled high copy number barcoded plasmids (*Ho et al., 2009*; *Magtanong et al., 2011*). At least 25,000 transformants were scraped from agar plates for fivefold replication of the library, and frozen stocks were made. All OE experiments were done in liquid YPD medium with G418 (200 mg/L) added for plasmid selection. Experiments interrogating single genes were performed via culture growths of yeast strains transformed with plasmids of interest, grown for 10 generations in YPD medium supplemented with G418 in shake flasks or test tubes at 30°C with shaking.

## Moby 2.0 competitive growth

The competition experiments were performed as previously described (*Ho et al., 2009*; *Magtanong et al., 2011*; *Piotrowski Jeff and Simpkins, 2015*). Briefly, frozen glycerol stocks of library transformed cells were thawed into 100 ml of liquid YPD with G418 (200 mg/L) at a starting $OD_{600}$ of 0.05. The remaining cells from the frozen stocks were pelleted by centrifugation and represented the starting pool (generation 0) for each strain. After precisely five generations, each pooled culture was diluted to an $OD_{600}$ of 0.05 in fresh YPD containing G418, to maintain cells in log phase. At 10 generations, cells were harvested and cell pellets were stored at −80°C.

## Library construction, sequencing, and analysis

Plasmids were recovered from each pool using QIAprep spin miniprep kits (Qiagen, Hilden, Germany) after pretreatment with 1 μl R-Zymolyase (Zymo Research, Irvine, CA) and ~100 μl of glass beads, with vortexing for 5 min. Plasmid barcodes were amplified using primers containing Illumina multiplex adaptors as described in *Magtanong et al., 2011*; *Piotrowski Jeff and Simpkins, 2015*. Barcodes from three biological replicates pooled and split across three lanes on an Illumina HiSeq Rapid Run with single end 100 bp reads. Sequencing generated a median of 7,570,975 reads per barcode. Read data are available in the Short Read Archive under accession number GSE171586.

## Moby normalization and analyses

We experimented with several normalization strategies, including TMM in the edgeR package (*Robinson et al., 2010*; *McCarthy et al., 2012*) and simple library size normalization, in which barcode reads were divided by the total barcode read count in the sample, multiplied by 1 million to rescale for edgeR analysis. The latter provided the most robust procedure with the fewest assumptions. To recapture genes that were clearly present in the starting pool but completely absent after 10 generation growth, we performed a data imputation: genes with at least 20 normalized read counts (>5th percentile of normalized reads) in all three replicates of the starting pool but missing reads from the end-point analysis received a pseudocount of 1 added to the barcode reads at 10 generations. Measured and imputed data were analyzed using edgeR version 3.22.1, using a linear model with generation (0 or 10) as a factor. Genes whose barcodes were significantly different after 10 generations of growth in each strain at an FDR<0.05 were taken as significant (*Benjamini and Hochberg, 1995*). Fitness scores were calculated by taking the ratio of normalized reads at

generation 10 divided by reads at generation 0 (*Supplementary file 4*). Hierarchical clustering was performed on the $\log_2$(fold change) in normalized fitness scores using Cluster 3.0 (*Eisen et al., 1998*) and visualized using Java TreeView (*Saldanha, 2004*). We considered if differences in statistical power could explain differences in the number of significant genes. *Figure 1B* shows that the biological replicates were highly reproducible. The mean correlation among replicates per strain was generally high (0.74–0.89), aside of three triplicated strains (Y2209, YJM1592, and YJM978). The correlation was lower for these strains (0.55–0.65) even though their replicates agreed well (*Figure 1B*); the apparently lower correlation is almost certainly driven by noise in the nearly negligible fitness changes (i.e., $\log_2$ values close to 0). Strains with nearly identical correlation across replicates showed very different numbers of significant genes (e.g., Y12 and YJM1273). Thus, differences in statistical power cannot explain the differences in significant genes across strains.

Functional and biophysical enrichments were assessed using Wilcoxon rank-sum tests for continuous data (e.g., gene length, # of SNPs, and % amino acid content) and Hypergeometric tests for categorical terms, taking as the background data set the total number of measured genes (except for strain-specific gene lists, in which the background data set was a list of insignificant genes in that strain with FDR>0.1 and measured in at least two of the biological replicates). Because gene lists are heavily overlapping, standard FDR calculations over-correct p-values. We therefore took a stringent p-value of $5 \times 10^{-4}$ as significant, but also cite FDR significance in data files. Genes involved in translation that were removed from several analyses are listed in *Supplementary file 2*. Functional and biophysical enrichments are available in *Supplementary file 6*. The background gene lists used for enrichments are available in *Supplementary file 8*.

## Determining copy number using quantitative PCR

We measured Moby 2.0 plasmid copy numbers in strains grown 10 generations in log-phase as described above. Plasmid DNA was extracted from frozen cell pellets using phenol/chloroform and ethanol precipitation, which recovers both plasmid and genomic DNA. Quantitative PCR experiments were conducted using a Roche LightCycler 480 II and Roche LightCycler 480 SYBR Green I Master SYBR-Green (Bio-Rad, Hercules, CA). Primers were designed to detect the KAN-MX resistance gene located on plasmids and genomic *TUB1* (control) (*Supplementary file 3*). $C_T$ values for each sample were measured in technical triplicate with all experiments done in greater than three biological replicates. The $C_T$ values for KAN were internally normalized to *TUB1* expressed from the genome and under an extreme constraint on copy number. KAN/TUB1 ratios measured for each isolate carrying the 2-μm plasmid were adjusted to BY4743 KAN/TUB1 ratios measured as an internal control in each experiment. Data were scaled to BY4743 values, which were adjusted relative to a KAN-MX marked CEN copy number measured in BY4743 (in the same way outlined above for Moby 2.0 plasmids).

## 2-μm copy number analysis

We determined the native copy number of the 2-μm gene, *REP1*, using publicly available DNA sequencing data for each strain (*Bergström et al., 2014*; *Hose et al., 2015*; *Strope et al., 2015*). (Note: BY4741 sequence was used instead of BY4743.) We mapped the sequencing data for each strain to a *S. cerevisiae* genome using BWA-MEM (version 0.7.12-r1039; *Li and Durbin, 2010*). Summed read counts for each gene were calculated by HT-Seq (version 0.6.0; *Anders et al., 2015*). Read counts were normalized using RPKM.

## Cloning

To express GFP, 343 bp upstream of *TEF1* (TEF$^{PROM}$) was PCR amplified and sewn to a PCR product capturing the GFP ORF and *ADH1* terminator (ADH1$^{TERM}$) taken from the Yeast GFP Clone Collection (Thermo Fischer Scientific). PCR product was transformed into yeast with linearized Moby 1.0 empty vector (*Ho et al., 2009*) and homologous recombinants were selected and verified by sequencing.

Moby 2.0 plasmids expressing *ERG26* and *LOC1* were isolated from *E. coli* using a Qiagen Spin Miniprep Kit. *ERG26* AND *LOC1* deletions were generated by site-directed mutagenesis. The first methionine codon of each ORF was mutated to TAG using quick-change cloning (*Wang and Malcolm, 1999*). All constructs were verified with Sanger sequencing.

## Western blot analysis

Yeast strains were grown in synthetic complete media to log phase (OD$_{600}$ ~0.4). CEN-GFP was monitored by Western blot analysis, loading OD-normalized cells in sample buffer and using rabbit anti-GFP (Abcam) and mouse anti-PGK1 (Abcam) as a loading control, and imaging on the Licor Odyssey Infrared Imager.

## Transcriptome profiling (RNA-Seq) and analysis

Yeast strains described in *Supplementary file 1* were grown in biological triplicate in rich YPD medium with G418 at 30°C with shaking, for three generations to an OD$_{600}$ ~0.5. Cultures were pelleted by centrifugation and flash frozen with liquid nitrogen and maintained at −80°C until RNA extraction. Total RNA was extracted by hot phenol lysis (*Gasch, 2002*), digested with Turbo DNase (Invitrogen) for 30 min at 37°C, and precipitated with 5 M lithium acetate for 30 min at −20°C. rRNA depletion was performed using the Ribo-Zero (Yeast) rRNA Removal Kit (Illumina, San Diego, CA) and libraries were generated according to the TruSeq Stranded Total RNA sample preparation guide (revision E). cDNA synthesis was performed using fragment prime finish mix (Illumina, San Diego, CA) and purified using Agencourt AMPure XP beads (Beckman Coulter, Indianapolis, IN). Illumina adaptors were ligated to DNA using PCR (10 cycles). The samples were pooled, resplit, and run across three lanes on an Illumina HiSeq 2500 sequencer, generating single-end 100 bp reads, with ~7,494,848 reads per sample. Data are available in GEO accession number GSE171585 and *supplementary file 5*.

Reads were processed using Trimmomatic version 0.3 (*Bolger et al., 2014*), and mapped to the S288c reference genome (version R64-1-1) with BWA-MEM (version 0.7.12-r1039; *Li and Durbin, 2010*). Read counts for each gene were calculated by HT-Seq (version 0.6.0; *Anders et al., 2015*). Differentially expressed genes were identified by *edgeR* (*Robinson et al., 2010*) using a linear model with strain background as a factor and paired replicates, identifying genes differentially expressed in each strain relative to the average of all strains using an FDR cutoff of 0.05 (*Benjamini and Hochberg, 1995*). Hierarchical clustering was performed by Cluster 3.0 (*Eisen et al., 1998*) and visualized using Java TreeView (*Saldanha, 2004*). There was a total of 4802 genes that were significant in at least one strain.

## Acknowledgements

The authors thank Peipei Wang and Shinhan Shiu for their input on data analyses, Auguste Dutcher for calculating IUPRED scores, and members of the Gasch Lab for useful feedback.

## Additional information

### Funding

| Funder | Grant reference number | Author |
|---|---|---|
| National Cancer Institute | R01CA229532 | James Hose<br>Adam Jochem<br>Audrey P Gasch |
| U.S. Department of Energy | DE-SC0018409 | Michael Place<br>Audrey P Gasch |
| National Institutes of Health | GT32GM007133 | DeElegant Robinson |
| National Human Genome Research Institute | 5T32HG002760 | DeElegant Robinson |

The funders had no role in study design, data collection and interpretation, or the decision to submit the work for publication.

### Author contributions

DeElegant Robinson, Formal analysis, Validation, Investigation, Writing - original draft; Michael Place, Software, Formal analysis; James Hose, Adam Jochem, Validation, Investigation; Audrey P

Gasch, Conceptualization, Formal analysis, Supervision, Funding acquisition, Methodology, Writing - review and editing

### Author ORCIDs
DeElegant Robinson (ID) https://orcid.org/0000-0002-0476-3908
Audrey P Gasch (ID) https://orcid.org/0000-0002-8182-257X

### Decision letter and Author response
Decision letter https://doi.org/10.7554/eLife.70564.sa1
Author response https://doi.org/10.7554/eLife.70564.sa2

## Additional files

### Supplementary files
• Supplementary file 1. Strains used in this study.

• Supplementary file 2. Translation-related genes. Genes annotated as involved in translation that were removed from analysis, where indicated in the text.

• Supplementary file 3. Primers used for quantitative PCR to measure plasmid abundances.

• Supplementary file 4. Moby Fitness Scores and gene lists. Tab 1: Unnormalized read counts for each strain. Tab 2: Library-size normalized and scaled read counts for each strain, see Materials and methods. Tab 3: Average ($\log_2$) change in fitness and Benjamini and Hochberg-corrected FDR as outputted by edgeR, without data imputation. Tab 4: Average ($\log_2$) change in fitness and Benjamini and Hochberg-corrected FDR as outputted by edgeR, using data in which some ratios had been imputed, see Methods for details. Tab 5: List of commonly deleterious genes. Tab 6: Strain-specific deleterious genes for each strain. Tab 7: Strain-specific beneficial genes for each strain.

• Supplementary file 5. RNA-seq read counts. Tab 1: Unnormalized reads counts per gene as outputted by HT-Seq. Tab 2: Average $\log_2$(expression ratio) comparing indicated strain versus the mean of all strains, followed by the FDR value, as outputted by edgeR and for each strain.

• Supplementary file 6. Moby Functional Enrichments. Enrichments for commonly deleterious genes or strain-specific genes, as indicated in each tab title. Quantitative data were scored by Wilcoxon rank-sum tests and categorical data were scored by Hypergeometric test, as described in the Materials and methods. Each column indicates the category, enrichment p-value(s), and either Bonferroni corrected p-value (p/number of tests) or the significance score (1=FDR<0.05, 0=FDR>0.05) using Benjamini and Hochberg ranking.

• Supplementary file 7. RNA-Seq functional enrichments. Functional enrichment of differentially expressed (d.e.) genes in each strain using Hypergeometric tests. Overlap between the query cluster and comparison cluster of GO and compiled categories is indicated with various p-values from Hypergeometric tests.

• Supplementary file 8. Background gene sets used for statistical tests. Tab 1: List of Moby genes measured in all three replications at generation 0 in at least one strain, minus the set of 431 common genes. This list was used as the background data set for Wilcoxon rank-sum tests analyzing common genes. Tab 2: List of Moby genes with no effect (FDR>0.1) in each strain, used for Wilcoxon rank-sum tests of strain-specific genes. Tab 3: List of Moby genes significant in at least one strain (FDR<0.05), used for Hypergeometric enrichment tests analyzing common genes. Tab 4: List of Moby genes significant in at least one strain (FDR<0.05) excluding the 431 common genes, used for Hypergeometric tests for strain-specific genes.

• Transparent reporting form

### Data availability
Barcode sequencing data are available in the Short Read Archive under accession number GSE171586. RNA-Seq data are available in GEO accession number GSE171585.

The following datasets were generated:

| Author(s) | Year | Dataset title | Dataset URL | Database and Identifier |
|---|---|---|---|---|
| Gasch AP, Place M | 2021 | Natural Variation in the Fitness Consequences of Gene Amplification in Wild Saccharomyces cerevisiae Isolates [Bar-seq] | https://www.ncbi.nlm.nih.gov/geo/query/acc.cgi?acc=GSE171586 | NCBI Gene Expression Omnibus, GSE171586 |
| Gasch AP, Place M | 2021 | Natural Variation in the Fitness Consequences of Gene Amplification in Wild Saccharomyces cerevisiae Isolates [RNA-seq] | https://www.ncbi.nlm.nih.gov/geo/query/acc.cgi?acc=GSE171585 | NCBI Gene Expression Omnibus, GSE171585 |

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
