## [Decision Letter]

**Acceptance summary:**

This study investigates the tolerance of CNV and its variation across different genetic backgrounds by using *Saccharomyces cerevisiae* as a model organism. Interestingly, the results show universal effects common to most of the genetic backgrounds, but also strain-specific effects of the gene over-expression. Together, these findings show how the effect and fitness cost of expression changes depends on both the affected gene as well as the general genetic background in which the expression change takes place.

**Decision letter after peer review:**

Thank you for submitting your article "Natural variation in the consequences of gene overexpression and its implications for evolutionary trajectories" for consideration by *eLife*. Your article has been reviewed by 2 peer reviewers, and the evaluation has been overseen by a Reviewing Editor and Patricia Wittkopp as the Senior Editor. The reviewers have opted to remain anonymous.

Essential Revisions:

1. Please provide a more detailed and careful description of replicates and statistics, as well as a comprehensive dataset with all raw fitness scores etc, (see detailed suggestions of both reviewers).

2. Please provide a critical discussion of possible confounding factors, including for example the effect of the vector. Where possible, please try to better separate the vector effects from the specific effects.

3. Consider including more controls and/or lowering the conclusions regarding the example of Trp genes.

*Reviewer #1 (Recommendations for the authors):*

In this manuscript, the authors focus on the tolerance of CNV and its variation across different genetic backgrounds by using *Saccharomyces cerevisiae* as a model organism. More precisely they measured the fitness costs of more than 4,000 over-expressed genes in 15 natural isolates. They found universal effects common to most of the genetic backgrounds but also strain-specific effects of the gene over-expression (OE).

The topic as well as the strategy are interesting. However, I see major issues preventing the publication of this study in its current state.

Here, a few points I would like to highlight:

– The data from biological triplicates are incredibly well correlated. How exactly was "biological" triplicates defined in this study? Were they all from the same initial frozen transformation pool and then cultured three times for 10 generations? Initial plasmid coverage can significantly impact the bc enrichment analyses. Looking at the data I don't think the authors have real biological triplicate data, e.g. the imputated values are usually also presented in all triplicates.

– What's the number of genes that were quantifiable across different strain backgrounds at T0? How does the initial bc frequency impact the number of significant OE genes in each strain?

– For the statistical tests the authors performed, although the test and p-value was indicated, the actual statistics are mostly absent. More information is needed for these tests, for example the exact number and identity of the background genes in all the hypergeometric tests.

– The authors discussed several models for strain specific OE phenotypes, however, they were all conjectures with overwhelming confounding effects from plasmid maintenance/tolerance differences, general protein over expressing tolerance etc. I think these parts should be more of a discussion than part of the results.

– The experiments done on the beneficial OE genes are inconclusive at best and again, too much confounding factors can be in the play here to draw any solid conclusions.

*Reviewer #2 (Recommendations for the authors):*

1) It would be important to show the full fitness score distributions, including neutral and beneficial OEs, in a supplementary figure.

2) Reproducibility of the screen should be provided for each strain. I also wonder how does the number of toxic genes differ between strains that show very similar measurement errors / statistical power? Can still large differences be observed?

3) It would be biologically insightful to attempt to statistically separate strain-specific OE effects from the strain-specific cost of carrying the empty Moby plasmid. Thus, identify strain-specific effects that would not be expected based on empty plasmid cost. These cases might better reflect physiological differences between strains.

4) In the analysis of tryptophan-enriched genes, it would be important to include, as a negative control, a few other genes that have similar functions but are not tryptophan-enriched.

---

## [Author Response]

Essential Revisions (for the authors):1. Please provide a more detailed and careful description of replicates and statistics, as well as a comprehensive dataset with all raw fitness scores etc, (see detailed suggestions of both reviewers).

As outlined in more detail below, we added raw and normalized read counts, along with the originally provided fitness effects (both with and without data imputation), provided a new table that has the background gene sets used in the different statistical tests (Supplementary File 5), and addressed reviewer comments below about reproducibility and statistical power. We also added clarifying statements to the Methods about statistical power and reproducibility (page 21).

2. Please provide a critical discussion of possible confounding factors, including for example the effect of the vector. Where possible, please try to better separate the vector effects from the specific effects.

We have provided additional analysis and critical discussion that some of the apparent strain-specific effects could be due to generalized strain sensitivity to the vector or stress conditions. These changes on are on pages 10, 11, 12, and 14-15.

3. Consider including more controls and/or lowering the conclusions regarding the example of Trp genes.

Our results validated the hypothesis that we set out to test, that “DBVPG1373 is sensitive to conditions that deplete tryptophan from the cell.” As per the editor’s suggestion, we tempered the conclusions and added a sentence stating that, “It is possible that this strain is more sensitive to any OE genes under these conditions, if the cellular system is taxed in the absence of tryptophan.”

Reviewer #1 (Recommendations for the authors):In this manuscript, the authors focus on the tolerance of CNV and its variation across different genetic backgrounds by using Saccharomyces cerevisiae as a model organism. More precisely they measured the fitness costs of more than 4,000 over-expressed genes in 15 natural isolates. They found universal effects common to most of the genetic backgrounds but also strain-specific effects of the gene over-expression (OE).The topic as well as the strategy are interesting.

We thank the reviewer for their positive feedback.

However, I see major issues preventing the publication of this study in its current state.Here, a few points I would like to highlight:– The data from biological triplicates are incredibly well correlated. How exactly was "biological" triplicates defined in this study? Were they all from the same initial frozen transformation pool and then cultured three times for 10 generations? Initial plasmid coverage can significantly impact the bc enrichment analyses. Looking at the data I don't think the authors have real biological triplicate data, e.g. the imputated values are usually also presented in all triplicates.

Our data are biological triplicates, in that the 10 generations of growth were done independently on different days. We transformed each strain to least 5-fold replication (i.e. 5X more colonies that the number of plasmids) with the same pool of plasmid DNA. The biological replicates were performed independently on separate days, by thawing the original replicated pool and growing precisely 10 generations. We went to great care to handle replicates as precisely as possible, owing to the high replication. Interestingly, in other experiments in our lab done during stress (not shown) the results are more variable across replicates, suggesting that not all conditions will give as high replication as we have produced here. Nonetheless, by all reasonable measures, these can be considered biological replicates. Data were imputed only for genes that were well measured at Generation 0 – data imputation was done for the most deleterious genes that reproducibly drop out of the population after outgrowth, which is why they are often imputed in multiple of the replicate Generation-10 samples.

– What's the number of genes that were quantifiable across different strain backgrounds at T0? How does the initial bc frequency impact the number of significant OE genes in each strain?

Excluding YPS606 which was done only in duplicate, the number of genes measured at Generation 0 (and thus used as input in edgeR) ranged from 3525 in YPS128 to 4072 in strain BY4743. There was no correlation between the number of genes measured at Generation 0 and the number of significant genes (R2 = 0.007). As outlined below, differences in statistical power cannot explain our results, since strains with nearly identical correlation among replicates show wildly different numbers of significant genes. Thus, while there are always subtle differences in statistical power, this cannot explain our main conclusion – that different genetic backgrounds show different sensitivities to gene OE, due to generalizable and gene-specific responses.

– For the statistical tests the authors performed, although the test and p-value was indicated, the actual statistics are mostly absent. More information is needed for these tests, for example the exact number and identity of the background genes in all the hypergeometric tests.

We were careful to document all the statistical methods as well as details about what gene sets were used a background for the various tests. We now provide an additional supplemental Supplementary File 5, which summarizes genes used as the background set including: all measured genes except common genes (used for Wilcoxon and Fisher tests analyzing the Common gene set), genes with no effect in each strain, i.e. those that were well measured but had no fitness effect (FDR > 0.01, used for tests analyzing strain-specific gene sets listed in Dataset 1), and the total set of genes significant in one or more strains, less the common genes (used for hypergeometric tests for strain-specific gene lists). All of the gene lists on which enrichment and functional assessment were done area already provided in various tabs in Dataset 1. These gene lists and our detailed descriptions provide all the necessary information to repeat all the statistical analyses.

– The authors discussed several models for strain specific OE phenotypes, however, they were all conjectures with overwhelming confounding effects from plasmid maintenance/tolerance differences, general protein over expressing tolerance etc. I think these parts should be more of a discussion than part of the results.

We appreciate the reviewer’s point. As outlined elsewhere in this response letter, we have now added more clarification to the text that some of the effects are due to strain sensitivity to the Moby vector, we provide a new analysis that shows that 60% of strain-specific genes in Y12 may be due to it’s sensitivity to the empty vector, and we provide more balanced presentation at multiple points in the revised manuscript.

– The experiments done on the beneficial OE genes are inconclusive at best and again, too much confounding factors can be in the play here to draw any solid conclusions.

In the manuscript, we discuss a set of genes whose OE is beneficial to a number of strains. Many of the OE plasmids cloned a centromere, suggesting a link to their beneficial effect. We went to considerable lengths to elucidate the mechanism and if the cloned CEN explains the fitness benefit; however, in the end our results were not conclusive. We made no claims about the mechanism in the paper. As this information may still be useful to others in the field, we opted to keep it in the manuscript without a strong focus or conclusions.

Reviewer #2 (Recommendations for the authors):1) It would be important to show the full fitness score distributions, including neutral and beneficial OEs, in a supplementary figure.

These plots are not particularly informative; however we direct the readers to Figure 1B which shows the colorized magnitudes and distributions.

2) Reproducibility of the screen should be provided for each strain. I also wonder how does the number of toxic genes differ between strains that show very similar measurement errors / statistical power? Can still large differences be observed?

As presented above, the replicate correlations are misleading – strains with the fewest genes of large effect (i.e. strains in which most log2 fitness effects are close to 0) have a reduced replicate correlation simply because the correlation is driven by noise / subtle variation in near-zero values. The distributions and replication is evident from Figure 1B, since we show all the biological replicates for each strain in the figure. As discussed above for two strains with nearly identically replication, there are widely different numbers of significant genes. Thus, differences in statistical power cannot explain our results.

3) It would be biologically insightful to attempt to statistically separate strain-specific OE effects from the strain-specific cost of carrying the empty Moby plasmid. Thus, identify strain-specific effects that would not be expected based on empty plasmid cost. These cases might better reflect physiological differences between strains.

This would be very difficult to do. We do not know the reason for strain sensitivity to the empty Moby 2.0 library, and thus it is hard to know how to correct for. We did add several clarifications to the text and cited that 60% of the genes meeting our strain-specific criteria in Y12 were shared with at least one other strain sensitive to the Moby 2.0 vector, raising the possibility that these are not really gene-specific responses but may represent general sensitivity of those strains to gene OE.

4) In the analysis of tryptophan-enriched genes, it would be important to include, as a negative control, a few other genes that have similar functions but are not tryptophan-enriched.

As per the guidelines of the editor, we have added a clarification that it is possible that this strain is sensitive to all OE genes in the absence of tryptophan.